# Research on Comprehensive Evaluation of Equipment for the Tea Processing Craft

**Song Mei, Qinghai Jiang and Zhiyu Song ***

Nanjing Institute of Agricultural Mechanization, Ministry of Agriculture and Rural Affairs, Nanjing 210014, China
* Correspondence: songzhiyu@caas.cn

**Abstract:** Chinese tea has a long history, and the development level of tea mechanization in this country is in the leading position internationally. Equipment for the tea processing craft is one of the most widely used types in China; indispensable technology is used for high-quality tea in the relevant tea processing equipment. The performance of tea processing equipment is always a focus when it is being used. Meanwhile, research evaluating the performance of tea processing equipment has been rarely reported. Thus, it is important to find a methodology to evaluate the performance of equipment for the tea processing craft. In this paper, a new method of rough set theory, as well as a radar chart for quality evaluation of tea processed by different tea processing crafts, is created. Firstly, the important indexes of processed tea are presented, including moisture, tea polyphenols, free amino acids, caffeine, and vitamin C. Then, these indexes for nine different types of processed tea based on different tea processing crafts are experimented and detected. Thirdly, rough set theory and radar chart are adopted to solving the weight of each index and establishing comprehensive evaluation diagram of all kinds of samples. Then, the merit of samples can be achieved. Meanwhile, the best tea processing equipment craft can be also obtained through comparisons between the sensory evaluation and the radar chart. It is concluded that sample 2, sample 3 and sample 4 belong to the best processing samples, and it is verified that the far-infrared radiation tea green removing machine has a significant influence on tea processing equipment craft. In addition, from the comprehensive analysis of green tea sample, not only tea quality but equipment craft of tea processing can be analyzed.

**Keywords:** processing equipment craft; tea green removing; sensory evaluation; rough set theory; radar chart

## 1. Introduction

Tea green-removing machines [1] are an important type of tea processing equipment. It is indispensable technology for destroying the tissue and oxidase activity of fresh leaves by high temperatures, repressing the enzymatic oxidation of tea polyphenols in fresh leaves, and preventing discoloration during drying. Meanwhile, the crafting of tea processing equipment has significant effects on tea quality. At present, with the development of tea processing technology and people's demand for high-quality tea, people not only pay attention to the external factors of tea [2], including the shape, soup color, aroma, taste, leaf bottom, etc., but also to the internal nutrition of the tea (such as moisture, tea polyphenols, free amino acids, caffeine, vitamin C, etc.). There are various methods of crafting tea processing equipment in China. However, the tea processed by different types of equipment is mainly evaluated by experts' subjective feelings; there is little objective evaluation of tea quality. Thus it is of great importance to research the comprehensive evaluation of tea quality in terms of the crafting of the processing equipment.

As the main method of the decision-making support system, research regarding comprehensive evaluation has been carried out in many fields, such as in the military, in agriculture, etc. [3]. The most commonly applied methods for comprehensive evaluation include rough set theory and the fuzzy optimization model. Ye Wei et al. utilized rough

set theory to reduce the evaluation index for crane performance, optimized the number of indexes, improved the efficiency of evaluation, and facilitated the development and operation of evaluation systems [4]. Tong Y.F. proposed energy-saving evaluation architecture for lifting machinery from the perspective of metal structure, transmission mechanisms, and electronic control [5]. Xiao H.R. proposed a comprehensive performance evaluation of the cultivation machinery used in tea gardens [3]. With the technological development occurring across the world, more and more new technologies have come into being, such as the most famous new IT ABCD (A: intelligent algorithms, B: big data analysis, C: cloud computing, D: data-driven). These new technologies greatly enhance scientific research by improving the effectiveness of the established methods, and also reduce their complexity, providing a set of usable paradigms for problem solving. Furthermore, intelligent algorithms are widely used in entire industries. The intelligent algorithms can be divided into three types: the regression algorithm, the classification algorithm, and the correlation algorithm. Regression algorithms are used to accurately predict future trends, such as support vector regression (SVR). Classification algorithms are used for classification, such as typical decision trees, etc. The regression algorithm and classification algorithm are widely used. In a comprehensive field evaluation, some intelligent algorithms, such as GA, BP neural network, etc., were used to model and solute the performance optimal selection [6].

The present work was carried out in order to assess tea quality based on different processing equipment, which can provide proof of the convenience of optimal tea processing equipment craft selection for farmers. In Section 2, an experiment to assess nine types of tea produced with different tea processing equipment types is described, which was carried out in Longtan forest farm, Liyang city, Jiangsu province, and its results were achieved through a professional inclusion detection institution. The main objective of this experiment was to test the energy-saving capability of each equipment and the main index values of tea content, which are convenient for the following analysis of tea quality characteristics. In Sections 3 and 4, rough sets and radar chart models are introduced into the comprehensive evaluation of different equipment crafts; rough set theory is utilized to determine the weight of performance index and radar chart is used for comprehensive analysis. This paper determines the weights of five indexes according to the method of calculating index weights in the rough set theory, and adopts the subjective assessment method of tea garden management experts to determine the weights of the five indexes. In Section 5, according to the results of the experiment summarized in Section 2, radar charts for nine types of tea processing equipment are drawn, and the weight of the quality index was calculated by means of the rough set theory. The tea quality of nine types of processing equipment is analyzed based on radar charts. Finally, according to the quality of the tea, the advantages of the tea processing equipment crafts will be confirmed.

## 2. Experiment Process and Results

On 7 August 2018, tea was processed in Longtan forest farm, Liyang city, Jiangsu province, with different types of processing equipment [1]. The main intent of this experiment is to test the energy-saving effect of each type of equipment as well as the main index values of tea content, which is convenient for the following analysis of tea quality characteristics. At the same time, it is beneficial to determine the optimal supporting equipment. The tea varieties, fresh leaf grade, and tea processing equipment used in the experiment are shown in Table 1.

**Table 1.** Setting of tea processing control experiment.

| Sample | Tea Varieties | Fresh Leaf Grade | Tea Processing Equipment Craft |
|---|---|---|---|
| One | | | microwave tea green removing and traditional carding |
| Two | | | far infrared tea green removing, traditional carding |
| Three | | | Microwave tea green removing and radiation carding |
| Four | | | far infrared tea green removing, radiation carding |
| Five | Fuding pekoe | Single bud, one bud one leaf | roller tea green removing and traditional carding |
| Six | | | roller tea green removing and radiation carding |
| Seven | | | green spreading machine, infrared tea green removing, traditional carding |
| Eight | | | green spreading machine, roller tea removing green, radiation carding |
| Nine | | | green spreading machine, microwave tea green removing, traditional carding |

As shown in Table 1, Fuding pekoe is a famous traditional craft tea made from white tea needles. White tea is a specialty of our country, produced in Fujian. In China, either single bud or one bud and one leaf are mainly processed for Fuding pekoe. Because of its remarkable color, smell, and shape, it was chosen as the processing object for the research carried out in this paper. Chinese tea can be divided into quality tea and bulk tea. Quality tea includes a single bud or one bud and one leaf, and a few good-quality types use one bud and two leaves. In addition, bulk tea mainly mixes all kinds of tea types for a high yield and average quality. Quality Fuding pekoe tea is chosen as the research object. Therefore, its tea grade includes single bud and one bud and one leaf. In addition, in equipment technology crafts two, four, and seven, a far-infrared thermal radiation tea green removing machine is used. Compared with the traditional tea green removing machine, its energy-saving effect and tea green-removing effect need to be further studied. An energy consumption test of three kinds of tea green-removing equipment is selected for the experiment. The specific test results are shown in Table 2.

**Table 2.** Record of the energy consumption test on three kinds of tea green-removing equipment.

| Equipment Model/Name | Fresh Leaf Grade | Power (kW) | Tea Green-Removing Weight (kg) | Preheating/Tea Green-Removing Time (s) | Energy Consumption (kW·h) |
|---|---|---|---|---|---|
| CSW-15 microwave tea green-removing machine | Single bud, one bud one leaf | 15 | 10 | 270/2273 | 9.5 |
| CST30 drum tea green-removing machine | | 13 | 10 | 1800/2328 | 14.4 |
| 6CFS-15 far infrared tea green removing machine | | 8.5 | 10 | 60/1641 | 3.9 |

It can be seen from the energy consumption data of three kinds of green removing test equipment that under the same processing time and fresh tea leaves, the production efficiency of the 6CFS-15far infrared tea green removing machine is the highest among the three models, and it only takes 1641s to accomplish the green removing work of 10 kg fresh leaves, which is about 40% higher than that of the CSW-15 microwave tea green removing machine and the CST30 drum tea green removing machine. Under the condition of the same workload, the energy consumption of 6CFS-15 far infrared tea green removing

machine is 3.9 kW, which is 58.9% and 73.3% less than that of CSW-15 microwave tea green removing machine and CST30 drum tea green removing machine respectively.

In order to obtain tea green removing effect of 6CFS-15 far infrared tea green removing machine and verify the importance of 6CFS-15 far infrared tea green removing machine in the finished tea processing equipment craft. As shown in Table 1, nine types of tea processing equipment crafts are listed. And nine kinds of tea samples are processed by different processing equipment crafts, then nine tea test samples are selected from the nine finished tea samples, each test sample is 200g, the tea samples are tested by the tea quality supervision, inspection and testing center of the ministry of agriculture and rural affairs, the detection method is as follows:

The sample extraction method is as follows: sampling and grinding according to GB/8302-2013 and GB/T 8303-2013. Weigh 0.5 g of tea sample into a conical flask, add 50 mL of boiling distilled water, immediately move it into a boiling water bath for leaching for 30 min, shake it every 10 min, immediately filter (absorbent cotton) to a 100 mL volumetric flask after leaching, and wash the residue twice with 50 mL of hot distilled water, cool to room temperature, and then fix the volume to 100 mL. Each sample is repeated three times.

The test methods for moisture, tea polyphenols, free amino acids, caffeine, and Vitamin C is respectively based on GB/8304-2013, GB/T8313-2008, GB/T8314-2013, GB/T8312-2013, and GB/T8312-2013. The test report is issued as shown in Table 3.

**Table 3.** Test report of tea contents.

| Sample | Moisture (%) | Tea Polyphenols (%) | Free Amino Acids (%) | Caffeine (%) | Vitamin C (%) |
|---|---|---|---|---|---|
| | | | Test Item | | |
| One | 11.4 | 18.8 | 2.6 | 5.1 | 0.033 |
| Two | 6.6 | 16.9 | 2.8 | 4.8 | 0.069 |
| Three | 5.5 | 19.8 | 2.8 | 5.0 | 0.1 |
| Four | 10.5 | 16.2 | 2.7 | 5.0 | 0.053 |
| Five | 7.8 | 20.2 | 2.3 | 4.8 | 0.11 |
| Six | 9.4 | 19.8 | 2.6 | 4.5 | 0.092 |
| Seven | 8.2 | 22.3 | 2.2 | 5.1 | 0.061 |
| Eight | 8.2 | 20.8 | 2.2 | 4.9 | 0.04 |
| Nine | 8.4 | 21.1 | 2.2 | 4.7 | 0.029 |

At the same time, the tea is evaluated by experts, and the results are shown in Table 4.

It can be seen from Table 3 that the contents of free amino acids, caffeine, and vitamin C in the nine tea samples are basically unchanged, while the contents of tea polyphenols fluctuated greatly. Among them, the tea polyphenols content of far-infrared radiation green removing processing is the lowest 16.2%, the tea polyphenols content of non-thermal radiation green removing processing is the highest 22.3%, the highest content of tea polyphenols decreased by 6%, and the tea polyphenols content was significantly reduced.

As shown in Table 4, it can be seen that the highest average score of tea processed by the fourth tea processing equipment craft is 91.1, which is 7.3% higher than that of the tea processed by the sixth tea processing equipment craft; thus, the lowest total score is 83.8. Meanwhile, the aroma score of the tea processed by far-infrared radiation is as high as 93, and all scores are above 90. Therefore, it is verified that the far-infrared radiation tea green-removing machine has a significant influence on tea processing equipment craft. The far-infrared radiation tea green-removing machine also has a remarkable effect on improving tea quality.

**Table 4.** Sensory evaluation report from experts.

| Number | Shape (25%) Comments | Score | Soup Color (10%) Comment | Score | Aroma (25%) Comment | Score | Taste (30%) Comment | Score | Leaf Bottom (10%) Comment | Score | Average Score |
|---|---|---|---|---|---|---|---|---|---|---|---|
| One | full bud (slightly leafed), milli, brownish-green, slightly yellow | 88 | yellow (slightly reddish) | 86 | still high | 88 | still stick | 87 | full bud (slightly with leaves), green slightly yellow | 90 | 87.7 |
| Two | Full bud (slightly leafed), profuse and green. | 90 | still green, still clear and bright | 90 | highly fresh, tender, and aromatic with chestnut fragrance. | 93 | still fresh | 90 | full bud (slightly leafed), bright and green | 90 | 90.8 |
| Three | full bud (slightly leafed), slightly curly, multi-milliliter, green and moist | 91 | light yellow | 88 | highly fresh, with chestnut fragrance | 92 | still thick | 88 | full bud (slightly leafed), bright green | 90 | 90.0 |
| Four | full bud (slightly leafed), slightly curly, multi-milliliter, fresh green and moist | 92 | light green, bright | 92 | higher fresh | 89 | thick and sweet (slight fire smell) | 92 | full bud (slightly leafed), bright, green | 91 | 91.1 |
| Five | full bud (slightly leafed), slightly flat, multi-milliliter, dark green | 88 | light yellow | 88 | fresh with slightly floral | 90 | still thick | 87 | full bud (slightly leafed), bright, green | 91 | 88.5 |
| Six | full bud (slightly leafed), green and bright, multi-milliliter, dark green, fresher and more moist | 90 | light yellow | 87 | fresh | 89 | still thick and slightly astringent | 86 | full bud (slightly leafed), bright, green. | 94 | 83.8 |
| Seven | loose strip, slightly flat, slightly dark | 86 | green, bright | 90 | fresh and floral | 91 | still thick | 87 | tender, thick, flower-shaped, and yellow-green | 87 | 87.7 |
| Eight | flower shape, slightly curly, yellow-green | 86 | green and bright | 90 | fresh and fragrant | 93 | mellow and fresh | 89 | still tender, flower shape, yellow-green. | 86 | 89.1 |
| Nine | loose strip, slightly flat, slightly dark yellow-green. | 84 | yellow-green | 87 | faint scent | 88 | still thick and slightly astringent. | 86 | tender, flower-shaped, yellow-green | 87 | 86.2 |

## 3. Rough Set Theory and Weight Determination Method

### 3.1. Rough Set Theory [7,8]

Compared with other methods to determine the weight, it needs experts' subjective judgment, and the weight is determined by rough set theory, without providing any prior information outside the objective data set, and only according to the existing design and application data. The weight of each index in the optimized index system is calculated by using the formula of attribute importance of rough set theory. The attribute importance of rough set theory is to compare the changes of system classification before and after removing an index. If the change is large, it means that the removed index has a large

attribute importance, that is, a large weight. On the contrary, if the change is small, it means that the removed index has a small attribute importance, that is, a small weight.

*3.2. Determination of Attribute Importance and Information System [9,10]*

**Definition 1.** *Let $K = (U, R)$ be a knowledge base and an equivalence relation, and let GD(R) be the granularity of knowledge $r \in R$, if*

$$GD(r) = \frac{|r|}{|U^2|} = \frac{|R|}{|U^2|} \tag{1}$$

**Definition 2.** *Let K = (U,r) be a knowledge base ($r \in R$ as an equivalence relation) and Dis(r) is definition from knowledge $r \in R$:*

$$Dis(r) = 1 - GD(r) \tag{2}$$

*suppose $S = (U, A, V, f)$ is an information system, $A = C \cup D$, $X \subseteq C$ is an attribute subset, and $x \in X$ is an attribute. Consider the degree of finality of x to X, i.e., the degree of resolution improvement after adding attribute X to x. The greater the degree of improvement, the more important x is to X.*

**Definition 3.** *Let $X \in C$ be an attribute subset and $X \in C$ be an attribute, and let the attribute importance of x to X be $Sig_X(x)$, which is defined as:*

$$Sig_X(x) = 1 - \frac{|X \cup \{x\}|}{|X|} \tag{3}$$

*where $|X|$ presents $|IND(X)|$, let $\frac{U}{|IND(X)|} = \frac{U}{X} = \{X_1, X_2, \cdots, X_n\}$, then*

$$|X| = |IND(X)| = \sum_{i=1}^{n} |X_i|^2$$

Comparing the method introduced above and the subjectively determined weight, the method is based on test data and scientific calculations, mining the test data to find an intrinsic relation in the importance of index data, therefore, it is more objective and can improve the accuracy of evaluation results.

*3.3. The Method of Comprehensively Determining the Weight*

In the information system $S = (U, A, V, f)$, each attribute in the conditional attribute set $C \in A$ has different influences on the value of the decision attribute, so they are given different weights. The final weight $I$ in this paper is composed by subjective and objective components, the objective of which is calculated by the above calculation method and denoted as $P$. And the subjective one is based on expert's experience and denoted by $Q$.

$$I = \alpha Q + (1 - \alpha)P(0 \leq \alpha \leq 1) \tag{4}$$

where $\alpha$ is an empirical factor, and $0 \leq \alpha \leq 1$ reflects decision-makers' preference for subjective weight and objective weight in the decision-making process. If it is lager, the decision makers pay more attention to the experience of expert. On the contrary, they pay more attention to objective weigh. The two limits of formula (2) indicate that only the influence of objective weight or subjective weight, respectively, is emphasized.

## 4. Radar Chart Model Establishment

The radar chart model comes from the system evolution analysis method of TRIZ theory. The mode and route of technological evolution are hot topics in TRIZ's theory

of technological evolution. According to statistics, at present, there are about 20 typical evolution modes and 350 evolution routes in academic circles, among which the most famous ones are the 6 evolution modes proposed by Fey V. and Rivin V., the 10 evolution modes proposed by Zlotin B. and Zusman A., and the 11 evolution modes proposed by Mann D. Among them, Mann D. studied the evolution model of technology system in detail, and its technology evolution model is full of systematic and structured characteristics. It is easy to operate [3], so it is increasingly favored in academic circles. In this paper, Mann D. and the concept of the evolutionary potential of technology are used to design a model to describe the performance of a technical system from a multiple-perspective radar chart.

In this paper, according to the test index, a radar chart model is established as shown in Figure 1, which can be illustrated as in Table 5.

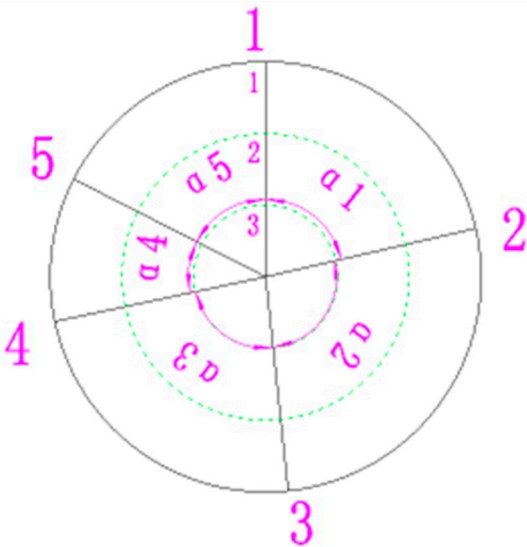

**Figure 1.** Radar chart of tea quality evaluation.

**Table 5.** Weight table of each index.

| Serial Number | 1 | 2 | 3 | 4 | 5 |
|---|---|---|---|---|---|
| index (I) | Moisture | Tea polyphenols | Free amino acids | Caffeine | Vitamin C |
| weight | $c_1$ | $c_2$ | $c_3$ | $c_4$ | $c_5$ |

As can be seen from Table 4, this paper solves five above indexes by the synthetic weights method based on rough set theory. It adopts the subjective assessment method of tea tasting experts to determine the weights of the above five indexes, where $\sum_{j=1}^{10} i_j = 1$. According to Figure 1, each axis direction is divided into three levels, and the position of each index on the axis is determined by the level of the evaluation index. The angle between axes is $\alpha$, $\alpha = 360° \times i$, which is determined by the index weight. To summarize, this method is used to determine the radar chart model and to quantitatively evaluate the tea quality characteristics.

## 5. Tea Quality Analysis

### 5.1. Determine the Weight of Each Index [5]

According to the index method above, a parameter system for evaluating tea quality is established as shown in Table 6.

**Table 6.** Evaluation parameter system of green tea quality.

| Target Layer | Evaluation Parameter | Mark |
|---|---|---|
| Comprehensive quality evaluation parameter system | Moisture (%) | $c_1$ |
| | Tea polyphenols (%) | $c_2$ |
| | Free amino acids (%) | $c_3$ |
| | Caffeine (%) | $c_4$ |
| | Vitamin C (%) | $c_5$ |

According to Tables 3, 5, and 6, the actual table of the parameters of information system attributes was established, as shown in Table 7.

**Table 7.** Green tea quality evaluation information system.

| U \ C | $c_1$ | $c_2$ | $c_3$ | $c_4$ | $c_5$ |
|---|---|---|---|---|---|
| $u_1$ | 11.4 | 18.8 | 2.6 | 5.1 | 0.033 |
| $u_2$ | 6.6 | 16.9 | 2.8 | 4.8 | 0.069 |
| $u_3$ | 5.5 | 19.8 | 2.8 | 5.0 | 0.1 |
| $u_4$ | 10.5 | 16.2 | 2.7 | 5.0 | 0.053 |
| $u_5$ | 7.8 | 20.2 | 2.3 | 4.8 | 0.11 |
| $u_6$ | 9.4 | 19.8 | 2.6 | 4.5 | 0.092 |
| $u_7$ | 8.2 | 22.3 | 2.2 | 5.1 | 0.061 |
| $u_8$ | 8.2 | 20.8 | 2.2 | 4.9 | 0.04 |
| $u_9$ | 8.4 | 21.1 | 2.2 | 4.7 | 0.029 |

In order to comprehensively evaluate the quality of tea, it is necessary to divide each index into three grades. In this system, grade 1 means "excellent", grade 2 means "good", and grade 3 means "average". The classification standard was determined by consulting the literature [11] and related data evaluating green tea quality, as shown in Table 8.

**Table 8.** Evaluation criteria of each index.

| Grade | Moisture | Tea Polyphenols | Free Amino Acids | Caffeine | Vitamin C |
|---|---|---|---|---|---|
| One | <5 | <16.5 | >2.5 | [2,5.5] | >0.05 |
| Two | [5,8.5] | [16.5,21] | (2,2.5) | [0,2], (5.5,7) | (0.04,0.05] |
| Three | >8.5 | >21 | [0,2] | >7 | [0,0.04] |

Comparing the data in Tables 7 and 8, an information system of tea quality can be obtained, as shown in Table 9.

**Table 9.** Information system for comprehensive evaluation of green tea quality.

| U \ C | $c_1$ | $c_2$ | $c_3$ | $c_4$ | $c_5$ |
|---|---|---|---|---|---|
| $u_1$ | 3 | 2 | 1 | 1 | 3 |
| $u_2$ | 2 | 2 | 1 | 1 | 1 |
| $u_3$ | 2 | 2 | 1 | 1 | 1 |
| $u_4$ | 3 | 1 | 1 | 1 | 1 |
| $u_5$ | 2 | 2 | 2 | 1 | 1 |
| $u_6$ | 3 | 2 | 1 | 1 | 1 |
| $u_7$ | 2 | 3 | 2 | 1 | 1 |
| $u_8$ | 2 | 2 | 2 | 1 | 3 |
| $u_9$ | 2 | 3 | 2 | 1 | 3 |

The weight of each index is calculated as follows:

$$X_1 = \{c_2, c_3, c_4, c_5\}$$

where $card(u_1) = card(u_2) = \cdots = card(u_9) = 1$.

Table 9 shows that $U/IND(X_1) = \{\{u_1\}, \{u_2, u_3, u_6\}, \{u_4\}, \{u_5\}, \{u_7\}, \{u_8\}, \{u_9\}\}$

$$\therefore |X_1| = 2^3 + 1 + 1 + 1 + 1 + 1 = 14$$

Because $U/IND(X) = \{\{u_1\}, \{u_2\}, \{u_3\}, \{u_4\}, \{u_5\}, \{u_6\}, \{u_7\}, \{u_8\}, \{u_9\}\}$

$$\therefore |X_1 \cup \{c_1\}| = 1^2 + 1^2 + 1^2 + 1^2 + 1^2 + 1^2 + 1^2 + 1^2 + 1^2 = 9$$

$$\therefore Sig_{X_1}(c_1) = 1 - \frac{|X_1 \cup \{c_1\}|}{|X_1|} = 1 - \frac{9}{14} = \frac{5}{14}$$

In the same way,

$$Sig_{X_2}(c_2) = 0.4, Sig_{X_3}(c_3) = \frac{5}{14}, Sig_{X_4}(c_4) = \frac{2}{11}, Sig_{X_5}(c_5) = \frac{4}{13}$$

According to Formula (2), the tea quality research experts gave subjective weights to each index, namely

$$Q_1 = 0.2, \ Q_2 = 0.3, \ Q_3 = 0.25, \ Q_4 = 0.1, \ Q_5 = 0.15$$

At the same time, the objective weight of each index was calculated as follows

$$P_{c1} = \frac{5/14}{5/14 + 0.4 + 5/14 + 2/11 + 4/13} \approx 0.223$$

Similarly, $P_{c2} \approx 0.249, P_{c3} \approx 0.223, P_{c4} \approx 0.113, P_{c5} \approx 0.192$.

According to the comprehensive weight determination method [12], this paper attached more importance to the proportion of objective weight. An empirical factor was selected, so the comprehensive weight of each attribute was [13,14]

$$I(c_1) = 0.4 \times 0.2 + 0.6 \times 0.223 = 0.2138$$
$$I(c_2) = 0.4 \times 0.3 + 0.6 \times 0.249 = 0.2694$$
$$I(c_3) = 0.4 \times 0.25 + 0.6 \times 0.223 = 0.2338$$
$$I(c_4) = 0.4 \times 0.1 + 0.6 \times 0.113 = 0.1078$$
$$I(c_5) = 0.4 \times 0.15 + 0.6 \times 0.192 = 0.1752$$

### 5.2. Establish Radar Chart

According to the comprehensive indexes [15] of the above five indexes, the proportion of each index in the radar chart could be determined. In addition, the position of each index in the axis of the radar chart was determined according to the grade division of Table 9. For the radar chart in Figure 1, $\alpha_1 = 0.2138 \times 360° \approx 77°$ was obtained according to the comprehensive weight index. In addition, we obtained $\alpha_2 \approx 97°, \alpha_3 \approx 84°, \alpha_4 \approx 39°, \alpha_5 \approx 63°$ in the same way. Therefore, a tea sample radar chart can be drawn using data from nine kinds of equipment technology.

### 5.3. Radar Chart Analysis of Comprehensive Evaluation of Sample Quality

The characteristics of the radar chart are the following: (1) the grades on the axis represent excellent, good, and average, respectively, and the position of the same index on the axis is different, which can distinguish the advantages and disadvantages of each tea index; (2) the larger the shadow area, the better the quality of the finished tea products.

In this paper, S1–S9 are used for sample 1–sample 9. According to the first characteristic, the indexes of each sample can be sorted. The evaluation of sample moisture content is S2 = S3 = S5 = S7 = S8 = S9 > S1 = S4 = S6; The evaluation of sample tea polyphenols is S4 > S1 = S2 = S3 = S5 = S6 = S8 > S7 = S9; The evaluation of free amino acids in samples is S1 = S2 = S3 = S4 = S6 > S5 = S7 = S8 = S9; The caffeine evaluation of the sample is the same for

nine samples, all of which reach the best; The evaluation of sample vitamin C is evaluated as S2 = S3 = S4 = S5 = S6 = S7 > S1 = S8 = S9. Through the grade evaluation of each index, we can only vaguely grasp the pros and cons of each index, but we can't judge the overall quality ranking of samples. The following calculation of sample shadow area can realize the ranking of samples.

According to the coordinates on the figures from Figure 2a to Figure 2i, the shadow area ratio of each radar chart is respectively 34.70%, 52.70%, 52.70%, 51.52%, 43.92%, 44.53%, 36.94%, 30.93% and 23.98%. According to the second characteristic, combined with shadow area, the area is sorted from large to small: S2 = S3 > S4 > S6 > S5 > S7 > S1 > S8 > S9. This sorting not only quantitatively distinguishes the advantages and disadvantages of samples, but also comprehensively expresses the quality characteristics of each sample, which is conducive to guiding the improvement of equipment and the adjustment of equipment technology.

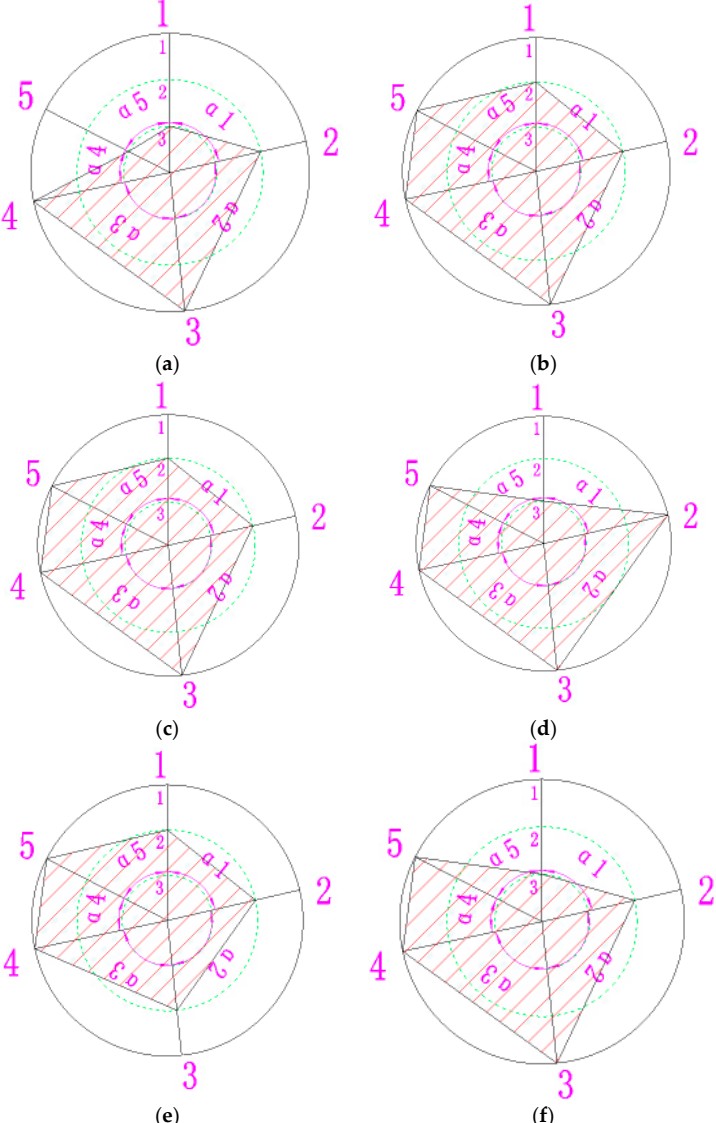

**Figure 2.** *Cont.*

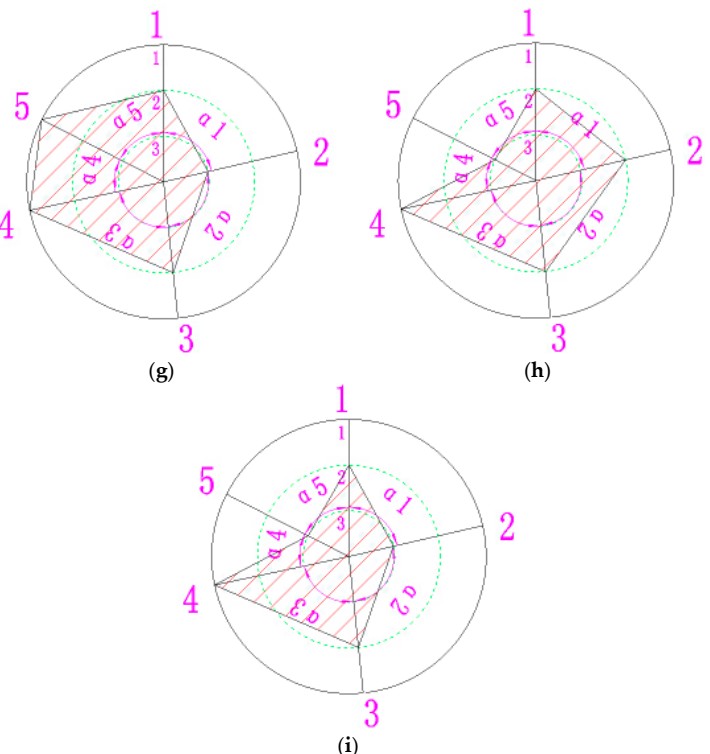

**Figure 2.** Comprehensive quality evaluation radar chart for nine samples.

*5.4. Comparative Analysis of Radar Image Evaluation and Sensory Evaluation and Inspection*

According to Table 4, the results of expert sensory evaluation are S4 > S2 > S3 > S8 > S5 > S7 = S1 > S9 > S6. The evaluation locates tea from the perspective of tea flavor quality, and the radar chart evaluation evaluates tea nutritional quality. Therefore, the two evaluation results are slightly different, but they can be re-evaluated according to the two evaluation results.

In addition, from the above evaluation results, it is concluded that sample 2, sample 3 and sample 4 belong to high-quality products, reflecting the good processing effect of the equipment corresponding to the samples. Among them, 6CFS-15 far infrared tea green removing machine was used in both crafts 2 and 4, it is verified that the far-infrared radiation tea green removing machine has a significant influence on tea processing equipment craft. On the other hand, it is also verified that the method of rough set theory and the radar chart for quality evaluation of processed tea by different tea processing crafts is reliable. Therefore, this method can not only evaluate the quality of tea, but also help researchers to determine the application value of developing tea processing equipment.

## 6. Conclusions

In this paper, the far-infrared radiation green-removing machine has not only an obvious energy-saving effect, but also a significant effect on reducing the bitterness and astringency of summer and autumn tea. And the highest content of tea polyphenols in tea is significantly reduced by 6%; The aroma of tea leaves after far-infrared radiation tea green removing machine and carding machine is improved by 5–10%, the quality of the tea leaves is obviously improved, and the overall sensory evaluation is improved by about 5%. The rough set theory is used to process the data on the nine green tea samples processed by equipment crafts, and the weight of each inclusion index is obtained. Firstly, the important indexes of processed tea are presented, including moisture, tea polyphenols, free amino acids, caffeine, and vitamin C. Then, these indexes for nine different types of processed tea based on different tea processing crafts are experimented and detected. Thirdly, the rough set theory and radar chart are adopted to solving the content weights

and establishing comprehensive evaluation diagrams of all the samples. Then, the merit of the samples can be achieved. Meanwhile, the best tea processing equipment craft can be also obtained through comparisons between the sensory evaluation and the radar chart. Combined with the radar chart model, the radar chart of comprehensive evaluation of the quality of nine kinds of green tea samples is established. And combined with the results of sensory evaluation, it is concluded that sample 2, sample 3 and sample 4 belong to the best processing samples, and their corresponding equipment and technology are better. Therefore, it is verified that the far-infrared radiation tea green removing machine has a significant influence on tea processing equipment craft and the method of rough set theory and the radar char is reliable. The evaluation method in this paper can also provide important reference for the evaluation of other food processing crafts.

**Author Contributions:** Conceptualization, S.M.; data curation, Z.S. and Q.J.; format analysis, S.M.; investigation, Z.S.; methodology, S.M.; validation, Z.S. and Q.J.; writing—original draft, S.M.; writing—review and editing, Z.S. All authors have read and agreed to the published version of the manuscript.

**Funding:** This work is supported by Yunan Province Key R&D Special Project (Grant No.202102AE090038); Promotion and Demonstration of Key Technologies for Fine Processing of Sichuan Tea (2022YFQ0110); Self-propelled intertillage fertilizer applicator for tea garden (2022C02010); The Fruit, Vegetable and Tea Harvesting Machinery Innovation Project of Chinese Academy of Agricultural Sciences.

**Institutional Review Board Statement:** Not applicable.

**Informed Consent Statement:** Not applicable.

**Data Availability Statement:** No new data were created or analyzed in this study. Data sharing is not applicable to this article.

**Conflicts of Interest:** The authors declare no conflict of interest.

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
