# Peer review of "Research on Comprehensive Evaluation of Equipment for the Tea Processing Craft"

_processes, doi:10.3390/pr11030778_

Round 1
Reviewer 1 Report
This manuscript is well written. It shows very good comprehensiveness in mathematics, algorithm , experimental settings, verification, etc., but there still exits some problems
1. In the introduction, the third paragraph, the serial numbers summarized in each Section are inaccurate.
2. In Section 2, the date of the exam is suggested to be accurate to the day.
3. Please add a comparative description of the three kinds of water-fixing machines in Table 2 and explain their functions in different processes.
4. Suggestions on technical terms of tea soup in Table 4 should revised.
5. The water-fixing machine has a great influence on the taste of tea. It is suggested that the author explain the specific relationship between the water-fixing machine and the fluctuation of tea polyphenols content from the inclusion test report(Table3). At the same time, combined with Table 4, the influence of green removing machine on expert sensory evaluation should be analyzed, which is convenient for comparison with the final evaluation and verifies the accuracy of comprehensive evaluation.
6. In conclusion, it is suggested to increase the influence of this method in other food processing fields.
7. In this paper, it is suggested to add references.
Author Response
All the modifies has been made, please track it in the manuscript
Reviewer 2 Report
This manuscript compares the different tea processing technology (equipment) and finally comments on the best possible technique from a quality perspective using rough set theory and radar data chart. Although this is a new study, the presentation lacks the flow of the article. Therefore, this manuscript needs to be revised drastically. Here I have highlighted a few points for the authors.
In the last part of the introduction, present the objective of the work properly. And then proceed to materials and methodology where you can incorporate different stages of your work. Please avoid section explanation in this part of the introduction.
Table 1 is not at all clear. How many varieties were used? Define the varieties. Although it is learnt that only nine samples were processed, Table 1 speaks differently, especially with leaf grades used. How many leaf grades were processed with each equipment technology? Please provide a clear presentation of Table 1.
In the sensory evaluation report, define the terms used such as Shang Gao, Shangnong, Gao Shuang, etc. In many instances scores are presented words, please avoid it.
Please use different kinds of numbering patterns for the Sections and Subsections throughout the manuscript.
Please incorporate the methods used for the determination of the different biochemical content in the methodology part and avoid the method name in the table.
Since tea quality is an unavoidable part of this work, the introduction demands a brief part of the quality prospect of processed tea in terms of biochemical content. In this regard, the authors may go through the article “catechin and caffeine content….” by Deka et. al. 2021 in the journal of food composition and analysis.
Please remove the horizontal and vertical inside borders of all the tables. I suggest the authors to look at the tables presented in that article (Deka et. al. 2021) and accordingly prepare the tables here.
Please state clearly whether the data presented in Table 1 is from a single analysis or the average of a number of analyses. All the analysis should be carried out at least in triplicate and the average data with standard error or deviation should be presented.
Line 38-39: Please add a reference
Line 32: correct the sentence.
Line 44: avoid an upper case in each word.
Please avoid repetition, for e.g. line 50-51 and line 61-62.
Line 66: the words “of tea trees” should be after “....fresh leaf grades”
Line 69: avoid an upper case in each word and please go through the whole manuscript.
Line 70: The sentence “ One sample..........spectrophotometer.” is not clear. Please rewrite it.
Line 75: Please change the caption of Table 3.....like –“Biochemical quality content of tea samples” or something like that.
In Table 3, please keep the units in parentheses.
Line no 124-132: Please provide reference.
Line no. 146: “This method will be described in detail later.”........ Please mention where it is.
Please club figures 2-10 into a single figure which will reduce repetitive captions. You can present it as Fig 2: (a), (b), .......
In the context of page 11, line no. 221-260, I suggest using codes for samples such as S1, S2, S3, ........ for sample 1, sample 2, sample 3,..... or something like that, starting from Table 1, which will be convenient and will increase the readability of the article.
In lines 221- 227, different types of sample numbers are used, what does it indicate?
In line 228, what is “Sample vitamin c”?
Line 239: Is it a heading or subheading......state clearly.
Please change the tenses to past wherever possible...... many a time there is a mixing of past and present. For example, sentences in line 47 and line 71.
Finally, there are scopes for improvement of the conclusion. The conclusion does not necessarily demands the results of each sample being analyzed. Moreover, rough set theory references [11, 12] should be avoided here.

Author Response

(The authors gave the same response as above.)

Reviewer 3 Report
The article describes the evaluation strategies for the tea quality of processing craft. Although the authors have covered all the aspects and explained them broadly, the following points need to be considered for the possible publication and modification of the present work.
1. The title is a bit confusing. Could you reframe the title?
2. Abstract is vaguer. Make it concise
3. English grammar and spell check is a must,
4. Table 4. shows sensory evaluation test report. What are the sensors?
5. Highlight the novelty of the proposed work.
6. Highlight the difference between the work already done in this field and the proposed work.
Author Response

(The authors gave the same response as above.)

Round 2
Reviewer 2 Report
Please give a specific reply to each point raised in the previous Review Report in a separate file.
It is difficult to track. Moreover, some suggestions are not incorporated. Please justify
Round 3
Reviewer 2 Report
What is “tea green removing machine”? Please write a sentence about it. To understand the very first phrase of your article, one will have to go through others’ work.
If your variety was “Fuding pekoe” then what is “Happiness pot white writing brush “? Why did not you revise it in the first revised version of the manuscript? Write an introductory sentence about your variety. There are so many popular varieties are there in China, what is the reason behind selecting this variety?
Why don’t you write clearly about leaf grades? It is not clear. How will the reader of your article go through? Is it “single bud and bud with first leaf” or what? It looks like you have processed tea with one fraction of the sample containing “only buds”, with another fraction containing “one leaf”. If you are not able to write it clearly, take help from seniors in this field.
In the table headings, please write “sample” instead of “number”. These are samples, not numbers.
If you give a number, say 3 for a heading, then for its subheading(s), you write 3.1, 3.2, and so on.
You are avoiding the methodology of the biochemical analysis part, As a common reader, if I want to design a work based on your findings (taking your work as a reference), how will I proceed? Even if you do not give the detail, you should provide a reference that will help readers to find the methodology used.
A single analysis of a sample is never accepted, there is the possibility of so many errors which include human error, instrumental error, sampling error, and many others and these errors are acceptable. That’s why one should carry out at least three replications and the average data should be presented.
I am not questioning the reliability of the Institution carrying out the analyses. Obviously, these esteemed Institutions are reliable. You just say how many replications were carried out, that’s all.
Please draw the first inside horizontal border of each table which will highlight your headings of the table. You did not follow the tables of the cited article (Deka et al. 2021) properly.
Line 114-118: Please try to re-write this part
Please remove the word “including” from line no. 36.
It is an international journal, and the article, if published, will be open to all, the international tea community should understand what you are contributing and presenting.
